# Determining Thrombogenicity: Using a Modified Thrombin Generation Assay to Detect the Level of Thrombotic Event Risk in Lupus Anticoagulant-Positive Patients

**DOI:** 10.3390/biomedicines11123329

**Published:** 2023-12-16

**Authors:** Pavla Bradáčová, Luděk Slavík, Jana Úlehlová, Eva Kriegová, Eliška Jará, Lenka Bultasová, David Friedecký, Jana Ullrychová, Jana Procházková, Antonín Hluší, Gayane Manukyan, Lenka Štefaničková

**Affiliations:** 1Department Clinical Hematology, Masaryk Hospital Ústí nad Labem, 40113 Ústi nad Labem, Czech Republic; eliska.jara@kzcr.eu (E.J.); jana.ullrychova@kzcr.eu (J.U.); 2Faculty of Medicine and Dentistry, Palacky University Olomouc, 77900 Olomouc, Czech Republic; 3Department of Hemato-Oncology, University Hospital Olomouc, Faculty of Medicine and Dentistry, Palacky University Olomouc, 77900 Olomouc, Czech Republic; jana.ulehlova@fnol.cz (J.Ú.); jana.prochazkova@fnol.cz (J.P.); antonin.hlusi@fnol.cz (A.H.); 4Department of Immunology, University Hospital Olomouc, Faculty of Medicine and Dentistry, Palacky University Olomouc, 77900 Olomouc, Czech Republic; eva.kriegova@fnol.cz (E.K.); gaya.manukyan@gmail.com (G.M.); 5Department Hematology and Biochemistry, University Hospital, 32300 Plzeň, Czech Republic; bultasoval@fnplzen.cz; 6Laboratory for Inherited Metabolic Disorders, University Hospital Olomouc, Faculty of Medicine and Dentistry, Palacky University Olomouc, 77900 Olomouc, Czech Republic; david.friedecky@upol.cz (D.F.); lenka.stefanickova@fnol.cz (L.Š.); 7Laboratory of Molecular and Cellular Immunology, Institute of Molecular Biology NAS RA, Yerevan 0014, Armenia

**Keywords:** antiphospholipid syndrome, antiphospholipid antibodies, thrombosis, thrombogenicity, thrombin generation assay, lupus anticoagulant

## Abstract

The aim of this study was to determine the thrombogenicity of lupus anticoagulant (LA) antibodies using a modified thrombin generation assay (TGA) with the addition of activated protein C (APC) in a group of 85 patients with LA-positive samples. Of these, 58 patients had clinical manifestations of antiphospholipid syndrome (APS) according to the Sydney criteria classification, i.e., each patient had thrombosis or foetal loss, and 27 patients did not show any clinical manifestations of APS. A comparison of the two groups’ TGA results revealed statistically significant differences (Fisher’s test *p* = 0.0016). The group of patients exhibiting clinical manifestations of APS showed higher thrombogenicity in 56.9% of patients, while the group of patients not yet exhibiting clinical manifestations of APS showed higher thrombogenicity in 25.9% of patients. There were no significant differences in the specificity of the TGA test between the groups of patients exhibiting similar clinical manifestations. Receiver operating characteristic curve analysis showed a more significant relationship (*p* = 0.0060) for TGA than for LA titre (*p* = 0.3387). These data suggest that the determination of LA thrombogenicity with the TGA assay leads to an increased prediction of the manifestation of a thromboembolic event. Our findings appear to be particularly relevant for the prediction of thrombotic events in patients with laboratory-expressed APS and no clinical manifestations.

## 1. Introduction

Together with anti-cardiolipin (aCL) and anti-β2-glycoprotein-I (anti-β2GPI), lupus anticoagulant (LA) belongs to a group of three so-called criterion antiphospholipid antibodies (aPL), the positive finding of which, according to the International Society on Thrombosis and Haemostasis (ISTH), is one of the defined laboratory criteria for the diagnosis of antiphospholipid syndrome (APS) [1,2,3,4,5,6].

In the case of aCL and anti-β2GPI, these are specific antibodies with known structures and known targets of action. However, this is not the case for the antibodies collectively referred to as LAs. In the case of LAs, it is not just one specific antiphospholipid antibody that could be detected directly with a specifically designed laboratory test [7]. LAs can include a set of diverse antiphospholipid antibodies that are characterised by laboratory prolonged coagulation tests dependent on low phospholipid concentrations [8,9]. These diverse pro-antibodies can be directed against many different targets, making it difficult to interpret their severity. Among these LAs, various so-called procoagulants can be included [10].

A laboratory evaluation of the manifestations of antiphospholipid antibodies is difficult. Detection of aPL can be performed using immunological, flowcytometric, molecular and functional methods. However, this detection does not allow the detection of the in vivo effect of aPL on target tissues. For this reason, it is always necessary to complete the clinical data of the patient [11]. Therefore, unfortunately, most studies are based on testing the clinical manifestations of APS in the context of proven specific antibodies. For example, Sciascia et al. reported in their work that the best diagnostic accuracy of APS is provided by the demonstration of LA positivity in combination with anti-β2GPI and anti-phosphatidylserine/prothrombin (anti-PS/PT) positivity [12]. This implies that only the strength of LA positivity, determined by conventional coagulation phospholipid-dependent tests, does not predict the risk of LA positivity for the manifestation of thromboembolic or arterial thrombotic events or pregnancy-related complications. 

For this reason, it is necessary to further identify the targets of action of the different types of positive lupoid antibodies, which may specifically act both in the cellular compartment (monocytes, platelets, and endothelial cells) and in the humoral compartment. 

Regarding the action of LA in the humoral compartment, earlier work has described the inhibitory activity of LA against activated protein C (APC) as one of the targets [13]. Thrombin activates protein C via the cofactor thrombomodulin. Protein C is a key inhibitor of haemocoagulation and regulates thrombus formation by inactivating FVa. In the case of APC modulation, haemostatic resistance to APC may increase, thereby increasing thrombin generation. Patients with LA positivity and APC resistance are frequently associated with thrombotic manifestations.

We focused our study on describing this activity by means of a modified thrombin generation assay (TGA).

The aim our study is to evaluate the hypercoagulable state in LA-positive patients with APS using TGA modified by the addition of activated protein C. We took advantage of the fact that TGA measures thrombin generation throughout the haemocoagulation process and thus provides a broader picture of the overall coagulation potential, since the basic coagulation tests prothrombin time, and activated partial thromboplastin time measure thrombin generation only during the initiation phase of coagulation. 

Similarly, Douxfils et al. [14] used this TGA test in their work, measuring the effect of hormonal contraceptives, the use of which can also cause hypercoagulability. Therefore, this modified TGA test could also be used to detect the sensitivity of different aPL to APC. Similarly, Radin et al. performed TGA in 108 patients with LA positivity and concomitant positivity of other aPLs and found the most severe findings with LA and TGA tests in patients who were tetrapositive (LA, anti-PS/PT, aβ2GPI, and aCL), and these patients also had significant clinical manifestations of APS [15].

TGA has also been used to verify the effect of anticoagulation therapy [16].

In a randomised trial, Cohen et al. compared TGA in 54 patients with APS who were being treated with warfarin and 56 patients who were being treated with rivaroxaban. Before treatment, both groups had similar endogenous thrombin potential (ETP) results, but after 42 days of treatment, ETP was significantly higher in the rivaroxaban group than in the warfarin group. In contrast, the warfarin group had a significantly prolonged lag phase and much lower peak thrombin generation than the rivaroxaban group [17].

### 1.1. β2-Glycoprotein-I

β2GPI appears to be a major target of various antiphospholipid antibodies [18]. β2GPI is composed of five domains (DI-DV) and occurs in two different conformations. More than 99% of β2GPI is found in the circular closed form, with DI associated with DV. In this form, target epitopes are hidden from aPL and complement binding. In the presence of aPL anti-β2GPI, the circular closed conformation changes to an open J-shaped conformation. In this conformation, DI epitopes are exposed and allow the binding of both aPL and complement [19,20].

In the body, β2GPI performs many functions necessary to maintain a stable haemostatic balance; through the binding of aPL to β2GPI, this balance can then be disrupted.

β2GPI also acts as an inhibitor of the inactivation of activated factor V with APC.

When bound to thrombin, β2GPI exerts a procoagulant effect, thereby preventing inactivation of thrombin by heparin cofactor I. By its antiplatelet effect, β2GPI inhibits the binding of platelets with von Willebrand factor, thereby reducing platelet adhesion [21]. Profibrinolytically, β2GPI exerts its binding to tissue plasminogen activator, acting as a cofactor of plasmin formation [22,23].

In the case of LA binding to β2GPI, its many functions may be disabled, thereby disrupting the haemostatic balance in favour of a prothrombotic state in patients with APS.

β2GPI is also expressed on the surface of placental trophoblast and maternal endo-body cells. The binding of various aPLs to β2GPI can induce a pro-inflammatory and antiangiogenic effect, which tends to cause foetal loss or preterm delivery in patients with APS [24].

### 1.2. Other Targets of Antiphospholipid Antibodies

Various aPLs can activate complement, whereby the C5b-9 complex is increased, and CD59 glycosyl phosphatidyl inositol (GPI) anchor expression is decreased, leading to the development of inflammation and subsequent thrombosis [25,26].

Another effect of aPL in the anti-β2GPI complex with β2GPI and/or annexin A2 is that aPL then activates neutrophils to form neutrophil extracellular traps (NETs), which may contribute to placental activation and thus induce preterm delivery [27].

High levels of NETs and DNA histones have been demonstrated in patients who are LA-positive and especially in patients who are triple-positive. Performing TGA showed increased thrombin generation in these patients [28,29].

The target of aPL may be the activation of platelets, which release increased thromboxane A2 in the presence of the β2GPI + anti-β2GPI + platelet factor 4 (PF4) complex [30].

### 1.3. Clinical Manifestation of Antiphospholipid Antibodies

The pathogenetic mechanisms by which aPLs induce disease manifestations in patients with APS are diverse, resulting in the extraordinary variability of clinical manifestations observed in these patients. Unfortunately, the molecular mechanisms that lead to the development of LAs and the expression of disease induced by them remain unclear.

In terms of antibody isotype, LAs with anticoagulant activity, as opposed to isolated aCL and anti-β2GPI, are the most important for the clinical manifestation of APS. LA positivity is a much higher risk factor for thromboembolism, cerebral ischemia, and recurrent reproductive loss compared with aCL and anti-β2GPI, as well as other non-criteria antibodies [6].

Choi et al. [31] studied 833 patients with recurrent aPL positivity and found that 96 patients exhibiting clinical manifestations of APS had LA positivity in 46.9% of cases, compared with a group of 737 asymptomatic carriers with LA positivity in only 25.6% of cases. For the other aPLs, there were no significant differences between the two groups. Mankee et al. and, similarly, Shaikhomar et al. demonstrated LA positivity as the strongest predictor of first-trimester pregnancy loss in their studies [32,33]. The results of a meta-analysis (27 relevant studies) conducted by Walter et al. confirm these facts: the highest risk for miscarriage, preeclampsia, or preterm birth is LA positivity or triple positivity [34]. In a cohort of 821 patients with systemic lupus erythematosus (SLE), Demir et al. demonstrated LA positivity as the highest risk marker for thrombosis, regardless of additional aPL positivity other than anti-β2GPI IgA [35]. Raj et al. demonstrated a significant association between LA positivity and haemolytic anaemia [36].

## 2. Materials and Methods

### 2.1. Study Design and Population

In our study, we included 85 LA-positive patients, some of whom were also aCL- and/or anti-β2GPI-positive; of these, 58 patients had clinical manifestations of APS, i.e., every patient had thrombotic or pregnancy complications; 27 patients had no clinical manifestations of APS to date.

All patients (85) were tested for criteria aPL: LA, aCL IgG, IgM, anti-β2GPI IgG, IgM, and anti--β2-glycoprotein-I domain I (anti-DI). All patients were LA-positive. Patients with heterozygous FV Leiden mutation, patients taking warfarin or heparin, were not included in the study. In patients treated with direct oral anticoagulants (DOACs), the drug was eliminated from the plasma under investigation using a DOAC stop tablet (Haematex Research, Hornsby, Australia) [37,38].

### 2.2. Sample Collection

Non-clotting blood was collected into tubes (BD Vacutainer ®, Franklin Lakes, New Jersey, USA) containing the anticoagulant sodium citrate 3.2%. The tubes were centrifuged no later than one hour after collection; for the preparation of platelet-poor plasma (PPP), double centrifugation at 2500× *g* was performed. Aliquots were then prepared and placed in a deep freezer at −80 °C. Prior to the actual measurements, the aliquots were thawed at 37 °C for 10 min.

### 2.3. Coagulation Tests for Lupus Anticoagulants

Determination of LA was performed based on the ISTH SSC international recommendation for the detection of LAs [1,39,40]. Detection is based on the ability of antiphospholipid antibodies present in the patient’s plasma to prolong the coagulation time in a phospholipid-dependent assay. The following tests were performed on a Sysmex CS 5100 coagulation analyser (Sysmex, Kobe, Japan) using reagents Diluted Russell Viper Venom Time (dRVVT) and aPTT (Siemens Healthineers, Erlangen, Germany). Measurements were performed according to the reagent kit manufacturer’s recommendation. For screening tests, we used reagents with low-concentration phospholipids and, for confirmation, reagents with high-concentration phospholipids [41] in the following three basic steps: 1, screening; 2, mixing test; and 3, confirmation [42].

The LA results were interpreted as positive/negative according to the ISTH SSC recommendations based on the calculation of the normalised ratio (NR). The calculation of the NR is as follows: NR = R screening (time patient/time polled normal plasma)/R confirmation (time patient/time polled normal plasma). The cut-off value for NR is 1.2. Samples that exceeded the cut-off >1.2 were determined to be LA-positive [43].

### 2.4. Chemiluminescence Immunoassays

Examination of aCL IgG, IgM, anti-β2GPI IgG, IgM, and anti-DI was performed using quantitative chemiluminescence immunoassays (CLIA) and using reagents QUANTA flash (Werfen, Barcelona, Spain) on a BioFlash analyser (Werfen, Barcelona, Spain) [44]. The principle of CLIA is the binding of aPL in the serum/plasma sample under investigation to paramagnetic particles coated with the appropriate surface. A compatible human Ig, labelled with isoluminol, binds to this resulting complex. Upon the addition of the trigger reagent, the chemiluminescence reaction is initiated [45]. During the chemiluminescence reaction, light is emitted and detected by the optical module in the relative light units (RLU) instrument. The measured RLUs are directly proportional to the concentration of each aPL in the sample. Through the 4PLC logistic curve, the measured RLUs are converted to chemiluminescence units (CU), and the cut-off recommended by the manufacturer is >20 U/mL [46].

### 2.5. Thrombin Generation Assay Modified by Activated Protein C

Thrombogenicity was determined by the thrombin generation assay using the Technothrombin^®^ TGA assay (Technoclone, Vienna, Austria) and APC reagent (Technoclone, Vienna, Austria) on a Ceveron alpha (Technoclone, Vienna, Austria). The TGA test can help to assess the risk of bleeding in bleeding coagulopathies or predict the risk of recurrent venous thromboembolism [47,48,49].

TGA measures the total haemostatic potential of plasma. The amount of thrombin is determined by the rate of cleavage of the fluorogenic substrate by thrombin. The rate of fluorescence is directly proportional to the amount of thrombin generated. The course of thrombin generation is shown in a curve in which the following four parameters are evaluated: tLag (min), time to the start of thrombin generation, reference range 2.3–5.1 min; tPeak (min), time from the start of thrombin generation to its maximum concentration, reference range 6.2–9.8 min.; mPeak (nM), maximum concentration of generated thrombin, reference range 21.7–214.8 nM; AUC (nM) = endogenous potential (ETP); total amount of generated thrombin, area under the curve, reference range 1.014–1.949 nM.

In our study, for each PPP sample, one TGA measurement was performed with a reagent with added activated protein C (+APC) and a second measurement without added activated protein C (−APC). From both measurements, the TGA −APC/TGA +APC ratio was calculated for each parameter [50]. Thrombin generation in the sample was initiated with 7.16 pM of the recombinant tissue factor (rTF) resuspended in 0.32 μM of phospholipid micelles (containing 2.56 μM of phosphatidylcholine and 0.64 μM of phosphatidylserine). For APC, activated protein C (16 nmol/L) was added.

The TGA assay was performed in two separate reactions with/without APC by modifying the putative antiphospholipid antibody target, which is activated protein C. To evaluate the effect of antibodies, the ratio of both determinations was used, which allowed us to evaluate the effect of antibodies on APC without separate effects of the basic coagulation state in individual patients.

In our recent study, we used 47 healthy volunteers as a negative control group and 21 patients with known hereditary thrombophilia with heterozygous mutation FV Leiden to determine the cut-off for the modified thrombin generation test because there is no existence of any cut-off of the evaluation of the risk of thrombogenicity. Although all TGA parameters were measured, the effects of antibodies on the TGA-modified assay were evident, especially in lag time and the total amount of thrombin generated (AUC). Significant differences between groups were found for the parameter AUC R = AUC − APC/AUC + APC. In patients with FV Leiden heterozygous, AUC R values ranged from 1.20 to 25.32. Based on these, a cut-off of ≤4.5 (90th percentile) was set for the AUC R parameter [51].

### 2.6. Statistics

Statistical analysis was performed using GraphPad Prism 9.0 software (GraphPad Software, San Diego, CA, USA). Data log transformation was used to normalise the original (tLag R). Data were visualised with scatter plots, including medians and quartiles. To calculate the statistical significance of the differences between the groups with or without clinical manifestation, the Fisher’s exact test was used, and the Mann–Whitney test was used to compare individual patients. The specificity and sensitivity of the dRVVT test for LA and the modified TGA were evaluated using the receiver operating characteristic curve (ROC).

## 3. Results

Our cohort consisted of 85 patients, 31 women and 54 men; all patients were positive for LA (dRVVT). We divided them into groups according to the positivity of each aPL: single positivity (65), double positivity (11), and triple positivity (9). In double positivity, five patients were positive for LA and aCL IgG, and six patients were positive for LA and anti-β2GPI IgG. Of these, 58 were patients exhibiting clinical manifestations of APS according to the Sydney criteria classification; that is, each patient had thrombosis or foetal loss, and 27 patients were still not exhibiting clinical manifestations of APS.

We further divided the 58 patients with clinical manifestations of APS according to whether they had venous or arterial thrombosis or foetal loss. Forty-four patients had venous thrombosis; six had arterial thrombosis; two had venous and arterial thrombosis; five had obstetric complications; one had venous and obstetric complications.

We further subdivided each group of patients with similar clinical manifestations of APS according to whether they achieved a TGA cut-off ≤4.5 on the TGA measurements. Individual data are shown in Table 1 and Table 2.

In the group of 58 patients exhibiting clinical manifestations of APS, 56.9% (33) of patients reached our cut-off of ≤4.5 for determining an increased risk of thrombogenicity. In the group of 27 patients still not exhibiting clinical manifestations of APS, only 25.9% (7) patients achieved the cut-off of ≤4.5. Individual data are shown in Figure 1.

In the group of 11 patients with APS double positivity, 45.5% (5) of patients reached our cut-off of ≤4.5 for determining an increased risk of thrombogenicity. In the group of 9 patients with APS triple positivity, only 33.3% (3) patients achieved the cut-off of ≤4.5.

In the group of patients with thrombotic APS, 55.8% (29) reached the cut-off, and in the group with pregnancy APS, 60% (3) reached the cut-off. In the group of patients with venous thrombosis, 54.8% (24) reached the cut-off, and in the group with arterial thrombosis, 66.8% (4) reached the cut-off.

The patients (85) in the cohort (exhibiting/not exhibiting clinical manifestation of APS) achieved LA titres (dRVVT) in the range of NR = 1.25–3.23. There were 70 patients with a weak LA titre NR = 1.2–1.5; 9 with a moderate LA titer NR = 1.5–2.0; and 6 with a strong LA titre NR > 2.00. Based on these results, we performed a correlation between the clinical manifestation of APS and the LA NR value. Another correlation was performed between the clinical manifestation of APS and the TGA AUC R. The results of the correlations are shown in Figure 2.

## 4. Discussion

The thrombogenicity of antiphospholipid antibodies is highly variable, and LA antibodies have the highest relative probability of causing thromboembolism. Several studies have shown that lupus anticoagulant positivity is a risk factor for thrombosis. In the general population, the risk of thrombosis increases with the number of positive antibodies [52]. Since the work evaluates the positivity of the lupus anticoagulant titre, we do not take into account other factors that can affect thrombogenicity, particularly the recurrence of thrombotic complications, but also the occurrence in unusual locations, as well as the difference in manifestation between venous and arterial thromboembolism [53]. Therefore, their detailed characterisation has been widely studied. Determining the effect of antiphospholipid antibodies on the TGA assay is a major challenge because antibodies are known to occupy phospholipid po-peaks, thereby prolonging in vitro coagulation tests but paradoxically creating a thrombophilic state [8]. One explanation may be the elimination of activated protein C, which also requires phospholipid surfaces for its activation.

Our current study follows up on our previous work in which we investigated the determination of thrombogenicity in patients with APS single, double, and triple positivity of various aPL including patients with SN-APS [51].

Here, we evaluated a cohort of 85 patients with LA positivity (dRVVT), of which 58 patients exhibited clinical manifestations of APS; 27 patients did not exhibit clinical manifestations of APS. The number of single-positive patients was 65, double-positive patients was 11, and triple-positive patients was 9. Of the double-positive cases, five patients were positive for LA and aCL IgG, and six patients were positive for LA and anti-β2GPI IgG.

In all 85 patients, we performed a modified TGA test with the addition of an APC reagent [54]. To assess thrombogenicity, we used the AUC R parameter, which appears to be the most beneficial [50], using our previously established cut-off of ≤4.5 [51].

By comparing the results of the TGA test between the two groups, we found statistically significant differences (Fisher’s test *p* = 0.0016). In the group of patients exhibiting clinical manifestations of APS, 56.9% of patients reached the cut-off, and in the group of patients who were not exhibiting clinical manifestations of APS, only 25.9% of patients reached the cut-off.

In the group of patients with manifest thrombotic events, the specificity of the TGA test was 55.8%, with pregnancy complications being 60%, venous thrombosis 54.8%, and arterial thrombosis 66.8%. There were no significant differences in the TGA test specificity between the groups of patients exhibiting similar clinical manifestations, and individual clinical manifestations of thrombotic APS did not differ.

In the diagnosis of LA since its introduction in 2000, the dRVVT test has had the highest predictive value and has improved the correlation with the incidence of thrombosis the most [55]. The fact that the demonstration of multiple positivity of different lupoid anticoagulants in a patient is the strongest predictor of non-obstetric thrombosis complications means that they can be used to assess the severity of thrombotic risk in patients [51]. Similar findings were reported in a study by de Groot et al. [56]; DVT occurred significantly more frequently in patients with LA positivity and anti-beta2GPI or anti-anti-prothrombin.

ROC analysis showed a more significant relationship (*p* = 0.0060) of TGA AUC R than that of the LA NR titre (*p* = 0.3387). These data suggest that the determination of aPL thrombogenicity with the TGA assay leads to an increased prediction of the manifestation of a thromboembolic event.

Our cohort consisted of patients (with/without a clinical manifestation of APS) with the LA dRVVT titre, ranging from an NR of 1.25 to 3.23. Patients with a weak titre NR of 1.2–1.5 were the most represented (70 patients), while a numerically smaller group consisted of patients with an intermediate titre NR of 1. 5–2.0 (9 patients) and with a strong NR titre > 2.00 (6 patients). Boeer et al. demonstrated in their study that thrombin generation is variable in patients with prolonged dRVVT [57], and we similarly found no correlation between the TGA and dRVVT titre in our study. This may imply that a strong LA positivity titre may not show increased thrombogenicity at all and vice versa. We explain this by the fact that there are probably other targets of lupoid antibody action than activated protein C, whose elimination by antiphospholipid antibodies causes thrombotic complications [51].

## 5. Conclusions

In the diagnosis of LA, the dRVVT test has the highest diagnostic value [55,58]. By combining the dRVVT test and a modified TGA test sensitive to one of the most potent inhibitory systems of APC coagulation, the diagnosis of APS can be further refined [15,57].

In terms of our cohort of patients who were LA-positive, in the group of patients exhibiting clinical manifestations of APS, high thrombogenicity was found in 56.9% of patients. In the group of patients not exhibiting clinical manifestations of APS, high thrombogenicity was found in 25.9% of patients. This finding appears to be particularly relevant for the prediction of thrombotic events in patients who have laboratory-expressed APS but are not exhibiting clinical manifestations because, according to current guidelines, anticoagulation therapy is not indicated for these patients [1] although they may be at a high risk of future thrombotic events [59,60,61].

The introduction of TGA into laboratory practice could be a useful tool to refine and speed up the diagnosis of APS, or to identify patients at risk, and to predict the manifestation of thrombotic complications. A subsequent initiation of antithrombotic therapy could lead to a reduction in the manifestation of thrombosis and pregnancy complications [62,63,64,65].

## Figures and Tables

**Figure 1 biomedicines-11-03329-f001:**
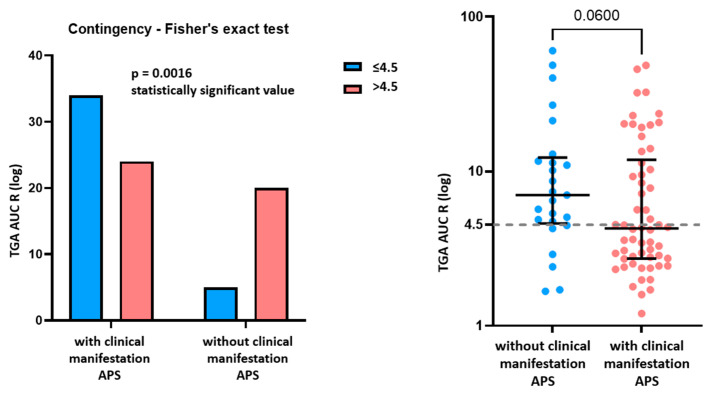
Visualisation of the representation of reaching the cut-off ≤4.5 in the groups of patients exhibiting/not exhibiting clinical manifestations of APS. TGA: thrombin generation assay; AUC: area under curve; R: ratio; APS: antiphospholipid syndrome.

**Figure 2 biomedicines-11-03329-f002:**
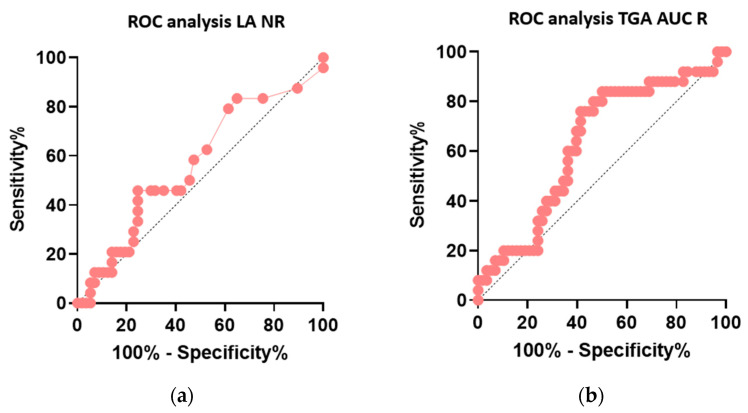
(**a**) ROC analysis sensitivity and specificity LA NR; (**b**) ROC analysis sensitivity and specificity of TGA AUC R. LA: lupus anticoagulant; NR: normalised ratio; TGA: thrombin generation assay; AUC: area under curve; R: ratio; APS: antiphospholipid syndrome. ROC: receiver operating characteristic curve.

**Table 1 biomedicines-11-03329-t001:** Overview of population and of the types of positivity and biomarkers in the examined patient cohort. LA: lupus anticoagulant; aCL: anti-cardiolipin; anti-β2GPI: anti-beta-2-glycoprotein I.

	N = 85
Sex (women/men)	31/54
Single-positivity LA	65
Double positivity	11
Positivity LA and aCL IgG	5
Positivity LA and anti-β2GPI IgG	6
Triple-positivity LA and aCL and anti-β2GPI	9

**Table 2 biomedicines-11-03329-t002:** Overview of clinical manifestations in the examined patient cohort. LA: lupus anticoagulant; TGA: thrombin generation assay; APS: antiphospholipid syndrome; DVT: deep vein thrombosis; PTE: pulmonary thromboembolism.

	LA Positive N = 85	TGA Positive, Cut-off ≤4.5/ TGA Negative, Cut-off >4.5
APS classification		
Thrombotic APS	52/85	29/23
Obstetric APS	5/85	3/2
Thrombotic and obstetric APS	1/85	1/0
Thrombotic APS	52/85	29/23
Venous thrombosis	44	24/20
Deep vein thrombosis	29	16/13
Pulmonary thromboembolism	7	4/3
PTE + DVT Portal vein thrombosis + DVT	7	4/3
	1	0/1
Arterial thrombosis	6	4/2
Cerebrovascular Event	6	4/2
Arterial and venous thrombosis	2	1/1
Myocardial infarction + DVT	1	0/1
Myocardial infarction + PTE	1	1/0
Obstetric APS	5/85	3/2
Foetal loss	5	3/2
Thrombotic and obstetric APS	1/85	1/0
Foetal loos + DVT	1	1/0
Without manifestations APS	27/85	7/20

## Data Availability

All data generated in this study are included in this published article.

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
