# Peer review of "Determining Thrombogenicity: Using a Modified Thrombin Generation Assay to Detect the Level of Thrombotic Event Risk in Lupus Anticoagulant-Positive Patients"

_biomedicines, 2023, doi:10.3390/biomedicines11123329_

Round 1

Reviewer 1 Report

Comments and Suggestions for Authors

Authors have addressed an interesting issue with the to better identify the risk of thrombosis in patients with APL antibodies.

They report that a modified thrombin generation assay better perform into the edentification of At risk APL carriers.

The report is interesting and results sounding but they needs, in my opinion, to be better presented after a more in-depth caractherization of the cohort.

- carriers of a double or triple positivity had different TGA results?

- were TGA results correlated with personal or clinical features (i.e. age, sex, number of cliical events, etc)?

- the manuscript should be rewritten, because it is too long in some sections (i.e. introduction), contains inappropriate sentences (methods) that can be reported as appropriate refs, ripetitive (i.e. discussion)

- fig 1 and 2 can be merged.

Author Response

  1. Carriers of a double or triple positivity had different TGA results?

We added a sentence on the line 314, page 7

In the group of 11 patients APS double positivity, 45.5 % (5) of patients reached our cut-off of ≤4.5 for determining increased risk of thrombogenicity. In the group of 9 patients APS triple positivity, only 33.3% (3) patients achieved the cut-off of ≤4.5.

However, in our study we did not look at double and triple positivity, but exclusively at patients with LA positivity as strongest marker of thrombosis.

  1. Were TGA results correlated with personal or clinical features (i.e. age, sex, number of clinical events, etc)?

Our findings showed no correlation between TGA AUC R and gender, age, or clinical manifestations. Clinical manifestations and TGA cut-off (AUC R) are shown in Table 2 in manuscript.

TGA cut-off ≤4,5: 19 female/21 male, 19 patient age <50, 19 patient age >50

TGA cut-off >4,5: 12 female/ 33 male, 20 patient age <50, 27 patient age >50

We added a sentence on the line 404, page 8:

Number of studies have shown that lupus anticoagulant are a risk factor for thrombosis. In the general population, the risk of thrombosis increases with the number of positive antibodies [52].

Durcan a M. Petri, „Chapter 2 - Epidemiology of the Antiphospholipid Syndrome", in Handbook of Systemic Autoimmune Diseases, roč. 12, R. Cervera, G. Espinosa, a M. Khamashta, Ed., in Antiphospholipid Syndrome in Systemic Autoimmune Diseases, vol. 12. , Elsevier, 2017, s. 17–30. doi: 10.1016/B978-0-444-63655-3.00002-8.

  1. The manuscript should be rewritten, because it is too long in some sections (i.e. introduction), contains inappropriate sentences (methods) that can be reported as appropriate refs, ripetitive (i.e. discussion)

We have rewritten introduction. The introduction of the manuscript was shortened by lines 6, when it was deleted.

  1. Fig 1 and 2 can be merged.

We merged its.

Reviewer 2 Report

Comments and Suggestions for Authors

The study involved 85 LA-positive patients. It compared TGA results between two groups: patients with clinical manifestations of APS (58 patients) and those without (27 patients). Significant differences in thrombogenicity were observed between the two groups. Higher thrombogenicity was found in 56.9% of patients with clinical APS manifestations, compared to 25.9% in patients without clinical manifestations. The specificity of the TGA test did not significantly differ between groups with similar clinical manifestations. The modified TGA assay appears to be a more effective tool for predicting thromboembolic events in patients with laboratory-expressed APS, particularly in those without clinical manifestations.

1. The study's sample size is relatively small, and its findings might benefit from validation in larger cohorts. The study focuses only on patients who are LA-positive, which might limit the applicability of the findings to the broader APS patient population.
2. As the study is observational, there may be confounding factors that were not controlled for or identified.

3. The introduction is too long to catch the importance of this study and may provide the mention of a modified TGA assay.

4. The conclusion is too long. The author may shorten the conclusion.

Author Response

  1. The study's sample size is relatively small, and its findings might benefit from validation in larger cohorts. The study focuses only on patients who are LA-positive, which might limit the applicability of the findings to the broader APS patient population.

We have added sentence on the line 417, page 8:

Our current study follows up our previous work in which we investigated the determination of thrombogenicity in patients with APS single, double, triple positivity of various aPL including SN-APS patients [51].

Bradáčová et al., „Determination of Thrombogenicity Levels of Various Antiphospholipid Antibodies by a Modified Thrombin Generation Assay in Patients with Suspected Antiphospholipid Syndrome", Int J Mol Sci, roč. 23, č. 16, s. 8973, srp. 2022, doi: 10.3390/ijms23168973.

  1. As the study is observational, there may be confounding factors that were not controlled for oridentified.

We have added sentence on the line 407 page 8:

Since the work evaluates the positivity of the lupus anticoagulant titer, we do not take into account other factors that can affect thrombogenicity, such as the recurrence of thrombotic complications in particular, but also the occurrence in unusual locations, as well as the difference in manifestation between venous and arterial thromboembolism [53].

Galli, D. Luciani, G. Bertolini, a T. Barbui, „Lupus anticoagulants are stronger risk factors for thrombosis than anticardiolipin antibodies in the antiphospholipid syndrome: a systematic review of the literature", Blood, roč. 101, č. 5, s. 1827–1832, bře. 2003, doi: 10.1182/blood-2002-02-0441.

  1. The introduction is too long to catch the importance of this study and may provide the mention of a modified TGA assay.

We have rewritten introduction. The introduction of the manuscript was shortened by lines 6, when it was deleted.

  1. The conclusion is too long. The author may shorten the conclusion.

 We have rewritten conclusion. The conclusion of the manuscript was shortened by lines 3, when it was deleted.

Round 2

Reviewer 1 Report

Comments and Suggestions for Authors

None.

Reviewer 2 Report

Comments and Suggestions for Authors

Thank you for the opportunity to review this interesting article. The author clearly response to the questions.